# CONVEX REGULARIZATION
# IN MONTE-CARLO TREE SEARCH

## ABSTRACT

Monte-Carlo planning and Reinforcement Learning (RL) are essential to sequential decision making. The recent AlphaGo and AlphaZero algorithms have shown how to successfully combine these two paradigms to solve large scale sequential decision problems. These methodologies exploit a variant of the well-known UCT algorithm to trade off the exploitation of good actions and the exploration of unvisited states, but their empirical success comes at the cost of poor sample-efficiency and high computation time. In this paper, we overcome these limitations by studying the benefit of convex regularization in Monte-Carlo Tree Search (MCTS) to drive exploration efficiently and to improve policy updates, as already observed in RL. First, we introduce a unifying theory on the use of generic convex regularizers in MCTS, deriving the first regret analysis of regularized MCTS and showing that it guarantees an exponential convergence rate. Second, we exploit our theoretical framework to introduce novel regularized backup operators for MCTS, based on the relative entropy of the policy update and on the Tsallis entropy of the policy. We provide an intuitive demonstration of the effect of each regularizer empirically verifying the consequence of our theoretical results on a toy problem. Finally, we show how our framework can easily be incorporated in AlphaGo and AlphaZero, and we empirically show the superiority of convex regularization w.r.t. representative baselines, on well-known RL problems across several Atari games.

## 1 INTRODUCTION

Monte-Carlo Tree Search (MCTS) is a well-known algorithm to solve decision-making problems through the combination of Monte-Carlo planning with an incremental tree structure (Coulom, 2006). Although standard MCTS is only suitable for problems with discrete state and action spaces, recent advances have shown how to enable MCTS in continuous problems (Silver et al., 2016; Yee et al., 2016). Most remarkably, AlphaGo (Silver et al., 2016) and AlphaZero (Silver et al., 2017b;a) couple MCTS with neural networks trained using Reinforcement Learning (RL) (Sutton & Barto, 1998) methods, e.g., Deep $Q$-Learning (Mnih et al., 2015), to speed up learning of large scale problems with continuous state space. In particular, a neural network is used to compute value function estimates of states as a replacement of time-consuming Monte-Carlo rollouts, and another neural network is used to estimate policies as a probability prior for the therein introduced PUCT action selection method, a variant of well-known UCT sampling strategy commonly used in MCTS for exploration (Kocsis et al., 2006). Despite AlphaGo and AlphaZero achieving state-of-the-art performance in games with high branching factor like Go (Silver et al., 2016) and Chess (Silver et al., 2017a), both methods suffer from poor sample-efficiency, mostly due to the polynomial convergence rate of PUCT (Xiao et al., 2019). This problem, combined with the high computational time to evaluate the deep neural networks, significantly hinder the applicability of both methodologies.

In this paper, we provide a unified theory of the use of convex regularization in MCTS, which proved to be an efficient solution for driving exploration and stabilizing learning in RL (Schulman et al., 2015; 2017a; Haarnoja et al., 2018; Buesing et al., 2020). In particular, we show how a regularized objective function in MCTS can be seen as an instance of the Legendre-Fenchel transform, similar to previous findings on the use of duality in RL (Mensch & Blondel, 2018; Geist et al., 2019; Nachum & Dai, 2020) and game theory (Shalev-Shwartz & Singer, 2006; Pavel, 2007). Establishing our theoretical framework, we can derive the first regret analysis of regularized MCTS, and prove that a generic convex regularizer guarantees an exponential convergence rate to the solution of the reg-

ularized objective function, which improves on the polynomial rate of PUCT. These results provide a theoretical ground for the use of arbitrary entropy-based regularizers in MCTS until now limited to maximum entropy (Xiao et al., 2019), among which we specifically study the relative entropy of policy updates, drawing on similarities with trust-region and proximal methods in RL (Schulman et al., 2015; 2017b), and the Tsallis entropy, used for enforcing the learning of sparse policies (Lee et al., 2018). Moreover, we provide an empirical analysis of the toy problem introduced in Xiao et al. (2019) to intuitively evince the practical consequences of our theoretical results for each regularizer. Finally, we empirically evaluate the proposed operators in AlphaGo and AlphaZero on problems of increasing complexity, from classic RL problems to an extensive analysis of Atari games, confirming the benefit of our novel operators compared to maximum entropy and, in general, the superiority of convex regularization in MCTS w.r.t. classic methods.

## 2 PRELIMINARIES

### 2.1 MARKOV DECISION PROCESSES

We consider the classical definition of a finite-horizon Markov Decision Process (MDP) as a 5-tuple $\mathcal{M} = \langle \mathcal{S}, \mathcal{A}, \mathcal{R}, \mathcal{P}, \gamma \rangle$, where $\mathcal{S}$ is the state space, $\mathcal{A}$ is the finite discrete action space, $\mathcal{R} : \mathcal{S} \times \mathcal{A} \times \mathcal{S} \to \mathbb{R}$ is the reward function, $\mathcal{P} : \mathcal{S} \times \mathcal{A} \to \mathcal{S}$ is the transition kernel, and $\gamma \in [0, 1)$ is the discount factor. A policy $\pi \in \Pi : \mathcal{S} \times \mathcal{A} \to \mathbb{R}$ is a probability distribution of the event of executing an action $a$ in a state $s$. A policy $\pi$ induces a value function corresponding to the expected cumulative discounted reward collected by the agent when executing action $a$ in state $s$, and following the policy $\pi$ thereafter: $Q^\pi(s, a) \triangleq \mathbb{E} \left[ \sum_{k=0}^\infty \gamma^k r_{i+k+1} | s_i = s, a_i = a, \pi \right]$, where $r_{i+1}$ is the reward obtained after the $i$-th transition. An MDP is solved finding the optimal policy $\pi^*$, which is the policy that maximizes the expected cumulative discounted reward. The optimal policy corresponds to the one satisfying the optimal Bellman equation (Bellman, 1954) $Q^*(s, a) \triangleq \int_\mathcal{S} \mathcal{P}(s'|s, a) \left[ \mathcal{R}(s, a, s') + \gamma \max_{a'} Q^*(s', a') \right] ds'$, and is the fixed point of the optimal Bellman operator $\mathcal{T}^* Q(s, a) \triangleq \int_\mathcal{S} \mathcal{P}(s'|s, a) \left[ \mathcal{R}(s, a, s') + \gamma \max_{a'} Q(s', a') \right] ds'$. Additionally, we define the Bellman operator under the policy $\pi$ as $\mathcal{T}_\pi Q(s, a) \triangleq \int_\mathcal{S} \mathcal{P}(s'|s, a) \left[ \mathcal{R}(s, a, s') + \gamma \int_\mathcal{A} \pi(a'|s') Q(s', a') da' \right] ds'$, the optimal value function $V^*(s) \triangleq \max_{a \in \mathcal{A}} Q^*(s, a)$, and the value function under the policy $\pi$ as $V^\pi(s) \triangleq \max_{a \in \mathcal{A}} Q^\pi(s, a)$.

### 2.2 MONTE-CARLO TREE SEARCH AND UPPER CONFIDENCE BOUNDS FOR TREES

Monte-Carlo Tree Search (MCTS) is a planning strategy based on a combination of Monte-Carlo sampling and tree search to solve MDPs. MCTS builds a tree where the nodes are the visited states of the MDP, and the edges are the actions executed in each state. MCTS converges to the optimal policy (Kocsis et al., 2006; Xiao et al., 2019), iterating over a loop composed of four steps:

1. **Selection:** starting from the root node, a *tree-policy* is executed to navigate the tree until a node with unvisited children, i.e. expandable node, is reached;

2. **Expansion:** the reached node is expanded according to the tree policy;

3. **Simulation:** run a rollout, e.g. Monte-Carlo simulation, from the visited child of the current node to the end of the episode;

4. **Backup:** use the collected reward to update the action-values $Q(\cdot)$ of the nodes visited in the trajectory from the root node to the expanded node.

The tree-policy used to select the action to execute in each node needs to balance the use of already known good actions, and the visitation of unknown states. The Upper Confidence bounds for Trees (UCT) sampling strategy (Kocsis et al., 2006) extends the use of the well-known UCB1 sampling strategy for multi-armed bandits (Auer et al., 2002), to MCTS. Considering each node corresponding to a state $s \in \mathcal{S}$ as a different bandit problem, UCT selects an action $a \in \mathcal{A}$ applying an upper bound to the action-value function

$$\text{UCT}(s, a) = Q(s, a) + \epsilon \sqrt{\frac{\log N(s)}{N(s, a)}}, \tag{1}$$

where $N(s, a)$ is the number of executions of action $a$ in state $s$, $N(s) = \sum_a N(s, a)$, and $\epsilon$ is a constant parameter to tune exploration. UCT asymptotically converges to the optimal action-value function $Q^*$, for all states and actions, with the probability of executing a suboptimal action at the root node approaching 0 with a polynomial rate $O(\frac{1}{t})$, for a simulation budget $t$ (Kocsis et al., 2006; Xiao et al., 2019).

## 3 REGULARIZED MONTE-CARLO TREE SEARCH

The success of RL methods based on entropy regularization comes from their ability to achieve state-of-the-art performance in decision making and control problems, while enjoying theoretical guarantees and ease of implementation (Haarnoja et al., 2018; Schulman et al., 2015; Lee et al., 2018). However, the use of entropy regularization is MCTS is still mostly unexplored, although its advantageous exploration and value function estimation would be desirable to reduce the detrimental effect of high-branching factor in AlphaGo and AlphaZero. To the best of our knowledge, the MENTS algorithm (Xiao et al., 2019) is the first and only method to combine MCTS and entropy regularization. In particular, MENTS uses a maximum entropy regularizer in AlphaGo, proving an exponential convergence rate to the solution of the respective softmax objective function and achieving state-of-the-art performance in some Atari games (Bellemare et al., 2013). In the following, motivated by the success in RL and the promising results of MENTS, we derive a unified theory of regularization in MCTS based on the Legendre-Fenchel transform (Geist et al., 2019), that generalizes the use of maximum entropy of MENTS to an arbitrary convex regularizer. Notably, our theoretical framework enables to rigorously motivate the advantages of using maximum entropy and other entropy-based regularizers, such as relative entropy or Tsallis entropy, drawing connections with their RL counterparts TRPO (Schulman et al., 2015) and Sparse DQN (Lee et al., 2018), as MENTS does with Soft Actor-Critic (SAC) (Haarnoja et al., 2018).

### 3.1 LEGENDRE-FENCHEL TRANSFORM

Consider an MDP $\mathcal{M} = \langle \mathcal{S}, \mathcal{A}, \mathcal{R}, \mathcal{P}, \gamma \rangle$, as previously defined. Let $\Omega : \Pi \to \mathbb{R}$ be a strongly convex function. For a policy $\pi_s = \pi(\cdot|s)$ and $Q_s = Q(s, \cdot) \in \mathbb{R}^{\mathcal{A}}$, the Legendre-Fenchel transform (or convex conjugate) of $\Omega$ is $\Omega^* : \mathbb{R}^{\mathcal{A}} \to \mathbb{R}$, defined as:

$$\Omega^*(Q_s) \triangleq \max_{\pi_s \in \Pi_s} \mathcal{T}_{\pi_s} Q_s - \tau \Omega(\pi_s), \tag{2}$$

where the temperature $\tau$ specifies the strength of regularization. Among the several properties of the Legendre-Fenchel transform, we use the following (Mensch & Blondel, 2018; Geist et al., 2019).

**Proposition 1** *Let $\Omega$ be strongly convex.*

- *Unique maximizing argument: $\nabla \Omega^*$ is Lipschitz and satisfies*

$$\nabla \Omega^*(Q_s) = \arg\max_{\pi_s \in \Pi_s} \mathcal{T}_{\pi_s} Q_s - \tau \Omega(\pi_s). \tag{3}$$

- *Boundedness: if there are constants $L_\Omega$ and $U_\Omega$ such that for all $\pi_s \in \Pi_s$, we have $L_\Omega \leq \Omega(\pi_s) \leq U_\Omega$, then*

$$\max_{a \in \mathcal{A}} Q_s(a) - \tau U_\Omega \leq \Omega^*(Q_s) \leq \max_{a \in \mathcal{A}} Q_s(a) - \tau L_\Omega. \tag{4}$$

- *Contraction: for any $Q_1, Q_2 \in \mathbb{R}^{\mathcal{S} \times \mathcal{A}}$*

$$\| \Omega^*(Q_1) - \Omega^*(Q_2) \|_\infty \leq \gamma \| Q_1 - Q_2 \|_\infty. \tag{5}$$

Although the Legendre-Fenchel transform $\Omega^*$ applies to every strongly convex function, for the purpose of this work we only consider a representative set of entropic regularizers.

### 3.2 REGULARIZED BACKUP AND TREE POLICY

In MCTS, each node of the tree represents a state $s \in \mathcal{S}$ and contains a visitation count $N(s, a)$. Given a trajectory, we define $n(s_T)$ as the leaf node corresponding to the reached state $s_T$. Let

$s_0, a_0, s_1, a_1..., s_T$ be the state action trajectory in a simulation, where $n(s_T)$ is a leaf node of $\mathcal{T}$. Whenever a node $n(s_T)$ is expanded, the respective action values (Equation 6) are initialized as $Q_\Omega(s_T, a) = 0$, and $N(s_T, a) = 0$ for all $a \in \mathcal{A}$. For all nodes in the trajectory, the visitation count is updated by $N(s_t, a_t) = N(s_t, a_t) + 1$, and the action-values by

$$Q_\Omega(s_t, a_t) = \begin{cases} r(s_t, a_t) + \gamma\rho & \text{if } t = T \\ r(s_t, a_t) + \gamma\Omega^*(Q_\Omega(s_{t+1})/\tau)) & \text{if } t < T \end{cases} \tag{6}$$

where $Q_\Omega(s_{t+1}) \in \mathbb{R}^\mathcal{A}$ with components $Q_\Omega(s_{t+1}, a), \forall a \in \mathcal{A}$, and $\rho$ is an estimate returned from an evaluation function computed in $s_T$, e.g. a discounted cumulative reward averaged over multiple rollouts, or the value-function of node $n(s_{T+1})$ returned by a value-function approximator, e.g. a neural network pretrained with deep $Q$-learning (Mnih et al., 2015), as done in (Silver et al., 2016; Xiao et al., 2019). We revisit the E2W sampling strategy limited to maximum entropy regularization (Xiao et al., 2019) and, through the use of the convex conjugate in Equation (6), we derive a novel sampling strategy that generalizes to any convex regularizer

$$\pi_t(a_t|s_t) = (1 - \lambda_{s_t})\nabla\Omega^*(Q_\Omega(s_t)/\tau)(a_t) + \frac{\lambda_{s_t}}{|\mathcal{A}|}, \tag{7}$$

where $\lambda_{s_t} = \epsilon|\mathcal{A}|/\log(\sum_a N(s_t,a)+1)$ with $\epsilon > 0$ as an exploration parameter, and $\nabla\Omega^*$ depends on the measure in use (see Table 1 for maximum, relative, and Tsallis entropy). We call this sampling strategy *Extended Empirical Exponential Weight* (E3W) to highlight the extension of E2W from maximum entropy to a generic convex regularizer.

### 3.3 CONVERGENCE RATE TO REGULARIZED OBJECTIVE

We show that the regularized value $V_\Omega$ can be effectively estimated at the root state $s \in \mathcal{S}$, with the assumption that each node in the tree has a $\sigma^2$-subgaussian distribution. This result extends the analysis provided in (Xiao et al., 2019), which is limited to the use of maximum entropy.

**Theorem 1** *At the root node $s$ where $N(s)$ is the number of visitations, with $\epsilon > 0$, $V_\Omega(s)$ is the estimated value, with constant $C$ and $\hat{C}$, we have*

$$\mathbb{P}(|V_\Omega(s) - V_\Omega^*(s)| > \epsilon) \le C\exp\{-\frac{N(s)\epsilon}{\hat{C}\sigma(\log(2 + N(s)))^2}\}, \tag{8}$$

where $V_\Omega(s) = \Omega^*(Q_s)$ and $V_\Omega^*(s) = \Omega^*(Q_s^*)$. From this theorem, we obtain that the convergence rate of choosing the best action $a^*$ at the root node, when using the E3W strategy, is exponential.

**Theorem 2** *Let $a_t$ be the action returned by E3W at step $t$. For large enough $t$ and constants $C, \hat{C}$*

$$\mathbb{P}(a_t \ne a^*) \le Ct\exp\{-\frac{t}{\hat{C}\sigma(\log(t))^3}\}. \tag{9}$$

## 4 ENTROPY-REGULARIZATION BACKUP OPERATORS

From the introduction of a unified view of generic strongly convex regularizers as backup operators in MCTS, we narrow the analysis to entropy-based regularizers. For each entropy function, Table 1 shows the Legendre-Fenchel transform and the maximizing argument, which can be respectively replaced in our backup operation (Equation 6) and sampling strategy E3W (Equation 7). Using maximum entropy retrieves the maximum entropy MCTS problem introduced in the MENTS algorithm (Xiao et al., 2019). This approach closely resembles the maximum entropy RL framework used to encourage exploration (Haarnoja et al., 2018; Schulman et al., 2017a). We introduce two novel MCTS algorithms based on the minimization of relative entropy of the policy update, inspired by trust-region (Schulman et al., 2015) and proximal optimization methods (Schulman et al., 2017b) in RL, and on the maximization of Tsallis entropy, which has been more recently introduced in RL as an effective solution to enforce the learning of sparse policies (Lee et al., 2018). We call these algorithms RENTS and TENTS. Contrary to maximum and relative entropy, the definition of the

Legendre-Fenchel and maximizing argument of Tsallis entropy is non-trivial, being

$$\Omega^*(Q_t) = \tau \cdot \text{spmax}(Q_t(s,\cdot)/\tau), \tag{10}$$

$$\nabla\Omega^*(Q_t) = \max\left(\frac{Q_t(s,a)}{\tau} - \frac{\sum_{a\in\mathcal{K}} Q_t(s,a)/\tau - 1}{|\mathcal{K}|}, 0\right), \tag{11}$$

where spmax is defined for any function $f : \mathcal{S} \times \mathcal{A} \to \mathbb{R}$ as

$$\text{spmax}(f(s,\cdot)) \triangleq \sum_{a\in\mathcal{K}} \left(\frac{f(s,a)^2}{2} - \frac{(\sum_{a\in\mathcal{K}} f(s,a) - 1)^2}{2|\mathcal{K}|^2}\right) + \frac{1}{2}, \tag{12}$$

and $\mathcal{K}$ is the set of actions that satisfy $1 + if(s,a_i) > \sum_{j=1}^{i} f(s,a_j)$, with $a_i$ indicating the action with the $i$-th largest value of $f(s,a)$ (Lee et al., 2018).

Table 1: List of entropy regularizers with Legendre-Fenchel transforms and maximizing arguments.

| Entropy | Regularizer $\Omega(\pi_s)$ | Legendre-Fenchel $\Omega^*(Q_s)$ | Max argument $\nabla\Omega^*(Q_s)$ |
|---|---|---|---|
| Maximum | $\sum_a \pi(a|s)\log\pi(a|s)$ | $\log\sum_a e^{\frac{Q(s,a)}{\tau}}$ | $\dfrac{e^{\frac{Q(s,a)}{\tau}}}{\sum_b e^{\frac{Q(s,b)}{\tau}}}$ |
| Relative | $D_{\text{KL}}(\pi_t(a|s)||\pi_{t-1}(a|s))$ | $\log\sum_a \pi_{t-1}(a|s)e^{\frac{Q_t(s,a)}{\tau}}$ | $\dfrac{\pi_{t-1}(a|s)e^{\frac{Q_t(s,a)}{\tau}}}{\sum_b \pi_{t-1}(b|s)e^{\frac{Q_t(s,b)}{\tau}}}$ |
| Tsallis | $\frac{1}{2}(\| \pi(a|s) \|_2^2 - 1)$ | Equation (10) | Equation (11) |

## 4.1 Regret analysis

At the root node, let each children node $i$ be assigned with a random variable $X_i$, with mean value $V_i$, while the quantities related to the optimal branch are denoted by $*$, e.g. mean value $V^*$. At each timestep $n$, the mean value of variable $X_i$ is $V_{i_n}$. The pseudo-regret (Coquelin & Munos, 2007) at the root node, at timestep $n$, is defined as $R_n^{\text{UCT}} = nV^* - \sum_{t=1}^{n} V_{i_t}$. Similarly, we define the regret of E3W at the root node of the tree as

$$R_n = nV^* - \sum_{t=1}^{n} V_{i_t} = nV^* - \sum_{t=1}^{n} \mathbb{I}(i_t = i)V_{i_t} = nV^* - \sum_i V_i \sum_{t=1}^{n} \hat{\pi}_t(a_i|s), \tag{13}$$

where $\hat{\pi}_t(\cdot)$ is the policy at time step $t$, and $\mathbb{I}(\cdot)$ is the indicator function.

**Theorem 3** *Let $\kappa_i = \nabla\Omega^*(a_i|s) + \frac{L}{p}\sqrt{\hat{C}\sigma^2 \log\frac{C}{\delta}/2n}$, and $\chi_i = \nabla\Omega^*(a_i|s) - \frac{L}{p}\sqrt{\hat{C}\sigma^2 \log\frac{C}{\delta}/2n}$, where $\nabla\Omega^*(.|s)$ is the policy with respect to the mean value vector $V(\cdot)$ at the root node $s$. For any $\delta > 0$, with probability at least $1 - \delta$, $\exists$ constant $L, p, C, \hat{C}$ so that the pseudo regret $R_n$ satisfies*

$$nV^* - n\sum_i V_i\left(\kappa_i + \frac{L}{p}\left(\frac{\tau(U_\Omega - L_\Omega)}{1-\gamma}\right)\right) \le R_n \le nV^* - n\sum_i V_i\left(\chi_i - \frac{L}{p}\left(\frac{\tau(U_\Omega - L_\Omega)}{1-\gamma}\right)\right).$$

This theorem provides bounds for the regret of E3W using a generic convex regularizer $\Omega$; thus, we can easily retrieve from it the regret bound for each entropy regularizer. Let $m = \min_a \nabla\Omega^*(a|s)$.

**Corollary 1** *Maximum entropy:*
$$nV^* - n\sum_i V_i\left(\kappa_i + L\left(\frac{\tau\log|A|}{1-\gamma}\right)\right) \le R_n \le nV^* - n\sum_i V_i\left(\chi_i - L\left(\frac{\tau\log|A|}{1-\gamma}\right)\right).$$

**Corollary 2** *Relative entropy:*
$$nV^* - n\sum_i V_i\left(\kappa_i + L\left(\frac{\tau(\log|A|-\frac{1}{m})}{1-\gamma}\right)\right) \le R_n \le nV^* - n\sum_i V_i\left(\chi_i - L\left(\frac{\tau(\log|A|-\frac{1}{m})}{1-\gamma}\right)\right).$$

**Corollary 3** *Tsallis entropy:*
$$nV^* - n\sum_i V_i\left(\kappa_i + \frac{L}{2}\left(\frac{|A|-1}{2|A|}\frac{\tau}{1-\gamma}\right)\right) \le R_n \le nV^* - n\sum_i V_i\left(\chi_i - \frac{L}{2}\left(\frac{|A|-1}{2|A|}\frac{\tau}{1-\gamma}\right)\right).$$

**Remarks.** The regret bound of UCT and its variance have already been analyzed for non-regularized MCTS with binary tree (Coquelin & Munos, 2007). On the contrary, our regret bound analysis in Theorem 3 applies to generic regularized MCTS. From the specialized bounds in the corollaries, we observe that the maximum and relative entropy share similar results, although the bounds for relative entropy are slightly smaller due to $\frac{1}{m}$. Remarkably, the bounds for Tsallis entropy become tighter for increasing number of actions, which translates in limited regret in problems with high branching factor. This result establishes the advantage of Tsallis entropy in complex problems w.r.t. to other entropy regularizers, as empirically confirmed by the positive results in several Atari games described in Section 5.

## 4.2 ERROR ANALYSIS

We analyse the error of the regularized value estimate at the root node $n(s)$ w.r.t. the optimal value: $\varepsilon_\Omega = V_\Omega(s) - V^*(s)$.

**Theorem 4** *For any $\delta > 0$ and generic convex regularizer $\Omega$, with some constant $C, \hat{C}$, with probability at least $1 - \delta$, $\varepsilon_\Omega$ satisfies*

$$-\sqrt{\frac{\hat{C}\sigma^2 \log \frac{C}{\delta}}{2N(s)}} - \frac{\tau(U_\Omega - L_\Omega)}{1-\gamma} \leq \varepsilon_\Omega \leq \sqrt{\frac{\hat{C}\sigma^2 \log \frac{C}{\delta}}{2N(s)}}. \tag{14}$$

To give a better understanding of the effect of each entropy regularizer in Table 1, we specialize the bound in Equation 14 to each of them. From (Lee et al., 2018), we know that for maximum entropy $\Omega(\pi_t) = \sum_a \pi_t \log \pi_t$, we have $-\log|\mathcal{A}| \leq \Omega(\pi_t) \leq 0$; for relative entropy $\Omega(\pi_t) = \mathrm{KL}(\pi_t||\pi_{t-1})$, if we define $m = \min_a \pi_{t-1}(a|s)$, then we can derive $0 \leq \Omega(\pi_t) \leq -\log|\mathcal{A}| + \log\frac{1}{m}$; and for Tsallis entropy $\Omega(\pi_t) = \frac{1}{2}(\| \pi_t \|_2^2 - 1)$, we have $-\frac{|\mathcal{A}|-1}{2|\mathcal{A}|} \leq \Omega(\pi_t) \leq 0$. Then,

**Corollary 4** *maximum entropy error:* $-\sqrt{\frac{\hat{C}\sigma^2 \log \frac{C}{\delta}}{2N(s)}} - \frac{\tau \log|A|}{1-\gamma} \leq \varepsilon_\Omega \leq \sqrt{\frac{\hat{C}\sigma^2 \log \frac{C}{\delta}}{2N(s)}}.$

**Corollary 5** *relative entropy error:* $-\sqrt{\frac{\hat{C}\sigma^2 \log \frac{C}{\delta}}{2N(s)}} - \frac{\tau(\log|A| - \log\frac{1}{m})}{1-\gamma} \leq \varepsilon_\Omega \leq \sqrt{\frac{\hat{C}\sigma^2 \log \frac{C}{\delta}}{2N(s)}}.$

**Corollary 6** *Tsallis entropy error:* $-\sqrt{\frac{\hat{C}\sigma^2 \log \frac{C}{\delta}}{2N(s)}} - \frac{|A|-1}{2|A|}\frac{\tau}{1-\gamma} \leq \varepsilon_\Omega \leq \sqrt{\frac{\hat{C}\sigma^2 \log \frac{C}{\delta}}{2N(s)}}.$

These results show that when the number of actions $|\mathcal{A}|$ is large, TENTS enjoys the smallest error; moreover, we also see that lower bound of RENTS is always smaller than for MENTS.

## 5 EMPIRICAL EVALUATION

In this section, we empirically evaluate the benefit of the proposed entropy-based MCTS regularizers. First, we complement our theoretical analysis with an empirical study of the synthetic tree toy problem introduced in Xiao et al. (2019), which serves as a simple scenario to give an interpretable demonstration of the effects of our theoretical results in practice. Second, we compare to AlphaGo and AlphaZero (Silver et al., 2016; 2017a), recently introduced to enable MCTS to solve large scale problems with high branching factor. Our implementation is a simplified version of the original algorithms, where we remove various tricks in favor of better interpretability. For the same reason, we do not compare with the most recent and state-of-the-art variant of AlphaZero known as MuZero (Schrittwieser et al., 2019), as this is a slightly different solution highly tuned to maximize performance, and a detailed description of its implementation is not available.

### 5.1 SYNTHETIC TREE

This toy problem is introduced in Xiao et al. (2019) to highlight the improvement of MENTS over UCT. It consists of a tree with branching factor $k$ and depth $d$. Each edge of the tree is assigned

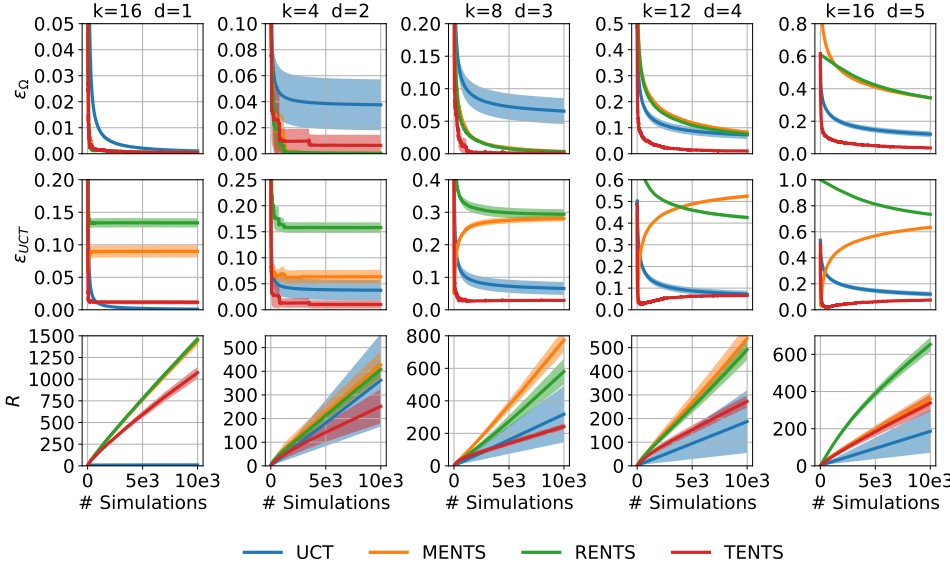

Figure 1: For each algorithm, we show the convergence of the value estimate at the root node to the respective optimal value (top), to the UCT optimal value (middle), and the regret (bottom).

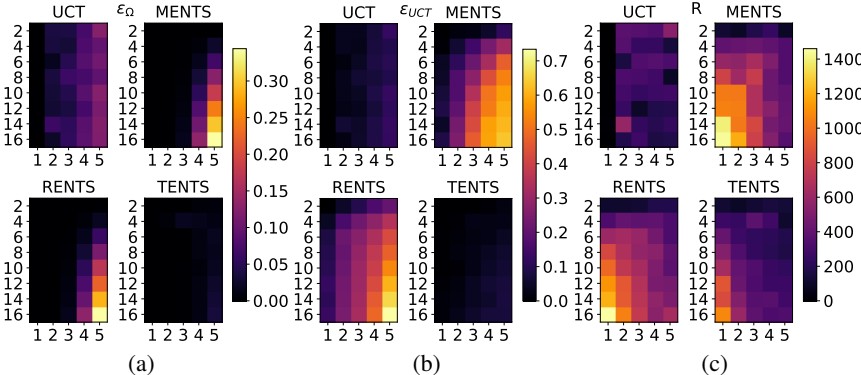

Figure 2: For different branching factor $k$ (rows) and depth $d$ (columns), the heatmaps show: the absolute error of the value estimate at the root node after the last simulation of each algorithm w.r.t. the respective optimal value (a), and w.r.t. the optimal value of UCT (b); regret at the root node (c).

a random value between $0$ and $1$. At each leaf, a Gaussian distribution is used as an evaluation function resembling the return of random rollouts. The mean of the Gaussian distribution is the sum of the values assigned to the edges connecting the root node to the considered leaf, while the standard deviation is $\sigma = 0.05$[1]. For stability, all the means are normalized between $0$ and $1$. As in Xiao et al. (2019), we create $5$ trees on which we perform $5$ different runs in each, resulting in $25$ experiments, for all the combinations of branching factor $k = \{2, 4, 6, 8, 10, 12, 14, 16\}$ and depth $d = \{1, 2, 3, 4, 5\}$, computing: (i) the value estimation error at the root node w.r.t. the regularized optimal value: $\varepsilon_\Omega = V_\Omega - V*$; (ii) the value estimation error at the root node w.r.t. the unregularized optimal value: $\varepsilon_{\text{UCT}} = V_\Omega - V*_{\text{UCT}}$; (iii) the regret $R$ as in Equation (13). For a fair comparison, we use fixed $\tau = 0.1$ and $\epsilon = 0.1$ across all algorithms. Figure 1 and 2 show how UCT and each regularizer behave for different configurations of the tree. We observe that, while RENTS and MENTS converge slower for increasing tree sizes, TENTS is robust w.r.t. the size of the tree and almost always converges faster than all other methods to the respective optimal value. Notably, the optimal value of TENTS seems to be very close to the one of UCT, i.e. the optimal value of the

---

[1] The value of the standard deviation is not provided in Xiao et al. (2019). After trying different values, we observed that our results match the one in Xiao et al. (2019) when using $\sigma = 0.05$.

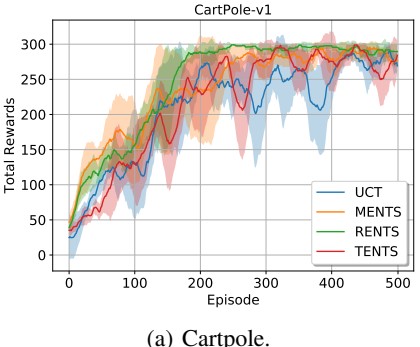 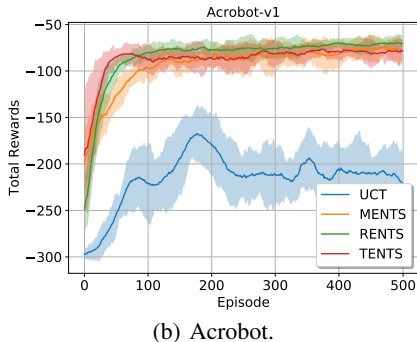

(a) Cartpole.          (b) Acrobot.

Figure 3: Cumulative rewards of AlphaZero with UCT and entropy-based operators, in CartPole (a) and Acrobot (b). Results are averaged over $5$ and $10$ seeds and show $95\%$ confidence intervals.

unregularized objective, and also converges faster than the one estimated by UCT, while MENTS and RENTS are considerably further from this value. In terms of regret, UCT explores less than the regularized methods and it is less prone to high regret, at the cost of slower convergence time. Nevertheless, the regret of TENTS is the smallest between the ones of the other regularizers, which seem to explore too much. These results show a general superiority of TENTS in this toy problem, also confirming our theoretical findings about the advantage of TENTS in terms of approximation error (Corollary 6) and regret (Corollary 3), in problems with many actions.

## 5.2 ENTROPY-REGULARIZED ALPHAZERO

In its standard form, AlphaZero (Silver et al., 2017a) uses the PUCT sampling strategy, a variant of UCT (Kocsis et al., 2006) that samples actions according to the policy

$$PUCT(s, a) = Q(s, a) + \epsilon P(s, a) \frac{\sqrt{N(s)}}{1 + N(s, a)}, \tag{15}$$

where $P$ is a prior probability on action selection, and $\epsilon$ is an exploration constant. A value network and a policy network are used to compute, respectively, the action-value function $Q$ and the prior policy $P$. We use a single neural network, with 2 hidden layers composed of 128 ELU units, and two output layer respectively for the action-value function and the policy. We run 500 AlphaZero episodes, where each episode is composed of 300 steps. A step consists of running 32 MCTS simulations from the root node, as defined in Section 2, using the action-value function computed by the value network instead of using Monte-Carlo rollouts. At the end of each cycle, the average action-value of the root node is computed and stored, the tree is expanded using the given sampling strategy, and the root node is updated with the reached node. At the end of the episode, a minibatch of 32 samples is built from the 300 stored action-values, and the network is trained with one step of gradient descent using RMSProp with learning rate $0.001$. The entropy-regularized variants of AlphaZero can be simply derived replacing the average backup operator, with the desired entropy function, and replacing PUCT with E3W using the respective maximizing argument and $\epsilon = 0.1$.

**Cartpole and Acrobot.** Figure 3 shows the cumulative reward of standard AlphaZero based on PUCT, and the three entropy-regularized variants, on the Cartpole and Acrobot discrete control problems (Brockman et al., 2016). While standard AlphaZero clearly lacks good convergence and stability, the entropy-based variants behave differently according to the problem. First, although not significantly superior, RENTS exhibits the most stable learning and faster convergence, confirming the benefit of relative entropy in control problems as already known for trust-region methods in RL (Schulman et al., 2015). Second, considering the small number of discrete actions in the problems, TENTS cannot benefit from the learning of sparse policies and shows slightly unstable learning in Cartpole, even though the overall performance is satisfying in both problems. Last, MENTS solves the problems slightly slower than RENTS, but reaches the same final performance. Although the results on these simple problems are not conclusive to assert the superiority of one method over the other, they definitely confirm the advantage of regularization in MCTS, and hint at the benefit of the use of relative entropy in control problems. Further analysis on more complex

Table 2: Average score in Atari over 100 seeds per game. Bold denotes no statistically significant difference to the highest mean (t-test, $p < 0.05$). Bottom row shows # no difference to highest mean.

|  | UCT | MaxMCTS | MENTS | RENTS | TENTS |
|---|---|---|---|---|---|
| Alien | **1,486.80** | **1,461.10** | **1,508.60** | **1,547.80** | **1,568.60** |
| Amidar | 115.62 | **124.92** | 123.58 | **125.58** | **121.84** |
| Asterix | 4,855.00 | **5,484.50** | **5,576.00** | **5,743.50** | **5,647.00** |
| Asteroids | 873.40 | 899.60 | 1,414.70 | 1,486.40 | **1,642.10** |
| Atlantis | 35,182.00 | **35,720.00** | **36,277.00** | 35,314.00 | **35,756.00** |
| BankHeist | 475.50 | 458.60 | **622.30** | **636.70** | **631.40** |
| BeamRider | **2,616.72** | **2,661.30** | **2,822.18** | 2,558.94 | **2,804.88** |
| Breakout | **303.04** | 296.14 | **309.03** | 300.35 | **316.68** |
| Centipede | 1,782.18 | 1,728.69 | **2,012.86** | **2,253.42** | **2,258.89** |
| DemonAttack | 579.90 | 640.80 | **1,044.50** | **1,124.70** | **1,113.30** |
| Enduro | **129.28** | 124.20 | 128.79 | **134.88** | **132.05** |
| Frostbite | 1,244.00 | 1,332.10 | **2,388.20** | **2,369.80** | **2,260.60** |
| Gopher | 3,348.40 | 3,303.00 | **3,536.40** | 3,372.80 | **3,447.80** |
| Hero | 3,009.95 | 3,010.55 | **3,044.55** | **3,077.20** | **3,074.00** |
| MsPacman | 1,940.20 | 1,907.10 | 2,018.30 | **2,190.30** | **2,094.40** |
| Phoenix | 2,747.30 | 2,626.60 | 3,098.60 | 2,582.30 | **3,975.30** |
| Qbert | 7,987.25 | 8,033.50 | 8,051.25 | 8,254.00 | **8,437.75** |
| Robotank | **11.43** | 11.00 | **11.59** | **11.51** | **11.47** |
| Seaquest | **3,276.40** | **3,217.20** | **3,312.40** | **3,345.20** | **3,324.40** |
| Solaris | 895.00 | 923.20 | **1,118.20** | **1,115.00** | **1,127.60** |
| SpaceInvaders | 778.45 | **835.90** | 832.55 | **867.35** | 822.95 |
| WizardOfWor | 685.00 | 666.00 | **1,211.00** | **1,241.00** | **1,231.00** |
| **# Highest mean** | 6/22 | 7/22 | 17/22 | 16/22 | **22/22** |

control problems will be desirable (e.g. MuJoCo (Todorov et al., 2012)), but the need to account for continuous actions, a non-trivial setting for MCTS, makes it out of the scope of this paper.

### 5.3 Entropy-regularized AlphaGo

The learning time of AlphaZero can be slow in problems with high branching factor, due to the need of a large number of MCTS simulations for obtaining good estimates of the randomly initialized action-values. To overcome this problem, AlphaGo (Silver et al., 2016) initializes the action-values using the values retrieved from a pretrained network, which is kept fixed during the training.

**Atari.** Atari 2600 (Bellemare et al., 2013) is a popular benchmark for testing deep RL methodologies (Mnih et al., 2015; Van Hasselt et al., 2016; Bellemare et al., 2017) but still relatively disregarded in MCTS. We use a Deep $Q$-Network, pretrained using the same experimental setting of Mnih et al. (2015), to initialize the action-value function of each node after expansion as $Q_{init}(s,a) = (Q(s,a) - V(s))/\tau$, for MENTS and TENTS, as done in Xiao et al. (2019). For RENTS we init $Q_{init}(s,a) = \log P_{\text{prior}}(a|s)) + (Q(s,a) - V(s))/\tau$, where $P_{\text{prior}}$ is the Boltzmann distribution induced by action-values $Q(s,.)$ computed from the network. Each experimental run consists of 512 MCTS simulations. The temperature $\tau$ is optimized for each algorithm and game via grid-search between 0.01 and 1. The discount factor is $\gamma = 0.99$, and for PUCT the exploration constant is $c = 0.1$. Table 2 shows the performance, in terms of cumulative reward, of standard AlphaGo with PUCT and our three regularized versions, on 22 Atari games. Moreover, we test also AlphaGo using the MaxMCTS backup (Khandelwal et al., 2016) for further comparison with classic baselines. We observe that regularized MCTS dominates other baselines, in particular TENTS achieves the highest scores in all the 22 games, showing that sparse policies are more effective in Atari. This can be explained by Corollary 6 which shows that Tsallis entropy can lead to a lower error at the root node even with a high number of actions compared to relative or maximum entropy.

## 6 Conclusion

We introduced a theory of convex regularization in Monte-Carlo Tree Search (MCTS) based on the Legendre-Fenchel transform. Exploiting this theoretical framework, we studied the regret of MCTS when using a generic strongly convex regularizer, and we proved that it has an exponential convergence rate. We use these results to motivate the use of entropy regularization in MCTS, particularly considering maximum, relative, and Tsallis entropy. Finally, we test regularized MCTS algorithms in discrete control problems and Atari games, showing its advantages over other methods.

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

## A RELATED WORK

Entropy regularization is a common tool for controlling exploration in Reinforcement Learning (RL) and has lead to several successful methods (Schulman et al., 2015; Haarnoja et al., 2018; Schulman et al., 2017a; Mnih et al., 2016). Typically specific forms of entropy are utilized such as maximum entropy (Haarnoja et al., 2018) or relative entropy (Schulman et al., 2015). This approach is an instance of the more generic duality framework, commonly used in convex optimization theory. Duality has been extensively studied in game theory (Shalev-Shwartz & Singer, 2006; Pavel, 2007) and more recently in RL, for instance considering mirror descent optimization (Montgomery & Levine, 2016; Mei et al., 2019), drawing the connection between MCTS and regularized policy optimization (Grill et al., 2020), or formalizing the RL objective via Legendre-Rockafellar duality (Nachum & Dai, 2020). Recently (Geist et al., 2019) introduced regularized Markov Decision Processes, formalizing the RL objective with a generalized form of convex regularization, based on the Legendre-Fenchel transform. In this paper, we provide a novel study of convex regularization in MCTS, and derive relative entropy (KL-divergence) and Tsallis entropy regularized MCTS algorithms, i.e. RENTS and TENTS respectively. Note that the recent maximum entropy MCTS algorithm MENTS (Xiao et al., 2019) is a special case of our generalized regularized MCTS. Unlike MENTS, RENTS can take advantage of any action distribution prior, in the experiments the prior is derived using Deep $Q$-learning (Mnih et al., 2015). On the other hand, TENTS allows for sparse action exploration and thus higher dimensional action spaces compared to MENTS. In experiments, both RENTS and TENTS outperform MENTS.

Several works focus on modifying classical MCTS to improve exploration. UCB1-tuned (Auer et al., 2002) modifies the upper confidence bound of UCB1 to account for variance in order to improve exploration. (Tesauro et al., 2012) proposes a Bayesian version of UCT, which obtains better estimates of node values and uncertainties given limited experience. Many heuristic approaches

based on specific domain knowledge have been proposed, such as adding a bonus term to value estimates (Gelly & Wang, 2006; Teytaud & Teytaud, 2010; Childs et al., 2008; Kozelek, 2009; Chaslot et al., 2008) or prior knowledge collected during policy search (Gelly & Silver, 2007; Helmbold & Parker-Wood, 2009; Lorentz, 2010; Tom, 2010; Hoock et al., 2010). (Khandelwal et al., 2016) formalizes and analyzes different on-policy and off-policy complex backup approaches for MCTS planning based on RL techniques. (Vodopivec et al., 2017) proposes an approach called SARSA-UCT, which performs the dynamic programming backups using SARSA (Rummery, 1995). Both (Khandelwal et al., 2016) and (Vodopivec et al., 2017) directly borrow value backup ideas from RL to estimate the value at each tree node, but they do not provide any proof of convergence.

## B  PROOFS

Let $\hat{r}$ and $r$ be respectively the average and the the expected reward at the leaf node, and the reward distribution at the leaf node be $\sigma^2$-sub-Gaussian.

**Lemma 1** *For the stochastic bandit problem E3W guarantees that, for $t \geq 4$,*

$$\mathbb{P}\big( \parallel r - \hat{r}_t \parallel_\infty \geq \frac{2\sigma}{\log(2+t)} \big) \leq 4|A| \exp\Big( - \frac{t}{(\log(2+t))^3} \Big).$$

**Proof 1** *Let us define $N_t(a)$ as the number of times action $a$ have been chosen until time $t$, and $\hat{N}_t(a) = \sum_{s=1}^t \pi_s(a)$, where $\pi_s(a)$ is the E3W policy at time step $s$. By choosing $\lambda_s = \frac{|A|}{\log(1+s)}$, it follows that for all $a$ and $t \geq 4$,*

$$\hat{N}_t(a) = \sum_{s=1}^t \pi_s(a) \geq \sum_{s=1}^t \frac{1}{\log(1+s)} \geq \sum_{s=1}^t \frac{1}{\log(1+s)} - \frac{s/(s+1)}{(\log(1+s))^2}$$

$$\geq \int_1^{1+t} \frac{1}{\log(1+s)} - \frac{s/(s+1)}{(\log(1+s))^2} ds = \frac{1+t}{\log(2+t)} - \frac{1}{\log 2} \geq \frac{t}{2\log(2+t)}.$$

*From Theorem 2.19 in Wainwright (2019), we have the following concentration inequality:*

$$\mathbb{P}(|N_t(a) - \hat{N}_t(a)| > \epsilon) \leq 2\exp\{-\frac{\epsilon^2}{2\sum_{s=1}^t \sigma_s^2}\} \leq 2\exp\{-\frac{2\epsilon^2}{t}\},$$

*where $\sigma_s^2 \leq 1/4$ is the variance of a Bernoulli distribution with $p = \pi_s(k)$ at time step $s$. We define the event*

$$E_\epsilon = \{\forall a \in \mathcal{A}, |\hat{N}_t(a) - N_t(a)| \leq \epsilon\},$$

*and consequently*

$$\mathbb{P}(|\hat{N}_t(a) - N_t(a)| \geq \epsilon) \leq 2|A| \exp(-\frac{2\epsilon^2}{t}). \tag{16}$$

*Conditioned on the event $E_\epsilon$, for $\epsilon = \frac{t}{4\log(2+t)}$, we have $N_t(a) \geq \frac{t}{4\log(2+t)}$. For any action $a$ by the definition of sub-gaussian,*

$$\mathbb{P}\Bigg( |r(a) - \hat{r}_t(a)| > \sqrt{\frac{8\sigma^2 \log(\frac{2}{\delta})\log(2+t)}{t}} \Bigg) \leq \mathbb{P}\Bigg( |r(a) - \hat{r}_t(a)| > \sqrt{\frac{2\sigma^2 \log(\frac{2}{\delta})}{N_t(a)}} \Bigg) \leq \delta$$

*by choosing a $\delta$ satisfying $\log(\frac{2}{\delta}) = \frac{1}{(\log(2+t))^3}$, we have*

$$\mathbb{P}\Bigg( |r(a) - \hat{r}_t(a)| > \sqrt{\frac{2\sigma^2 \log(\frac{2}{\delta})}{N_t(a)}} \Bigg) \leq 2\exp\Big( - \frac{1}{(\log(2+t))^3} \Big).$$

*Therefore, for $t \geq 2$*

$$\mathbb{P}\left(\| r - \hat{r}_t \|_\infty > \frac{2\sigma}{\log(2+t)}\right) \leq \mathbb{P}\left(\| r - \hat{r}_t \|_\infty > \frac{2\sigma}{\log(2+t)}\bigg| E_\epsilon\right) + \mathbb{P}(E_\epsilon^C)$$

$$\leq \sum_k \left(\mathbb{P}\left(|r(a) - \hat{r}_t(a)| > \frac{2\sigma}{\log(2+t)}\right) + \mathbb{P}(E_\epsilon^C) \leq 2|A|\exp\left(-\frac{1}{(\log(2+t))^3}\right)\right)$$

$$+ 2|A|\exp\left(-\frac{t}{(\log(2+t))^3}\right) = 4|A|\exp\left(-\frac{t}{(\log(2+t))^3}\right).$$

**Lemma 2** *Given two policies $\pi^{(1)} = \nabla\Omega^*(r^{(1)})$ and $\pi^{(2)} = \nabla\Omega^*(r^{(2)}), \exists L$, such that*

$$\| \pi^{(1)} - \pi^{(2)} \|_p \leq L \| r^{(1)} - r^{(2)} \|_p .$$

**Proof 2** *This comes directly from the fact that $\pi = \nabla\Omega^*(r)$ is Lipschitz continuous with $\ell^p$-norm. Note that $p$ has different values according to the choice of regularizer. Refer to Niculae & Blondel (2017) for a discussion of each norm using Shannon entropy and Tsallis entropy regularizer. Relative entropy shares the same Properties with Shannon Entropy.*

**Lemma 3** *Consider the E3W policy applied to a tree. At any node $s$ of the tree with depth $d$, Let us define $N_t^*(s,a) = \pi^*(a|s).t$, and $\hat{N}_t(s,a) = \sum_{s=1}^t \pi_s(a|s)$, where $\pi_k(a|s)$ is the policy at time step $k$. There exists some $C$ and $\hat{C}$ such that*

$$\mathbb{P}\left(|\hat{N}_t(s,a) - N_t^*(s,a)| > \frac{Ct}{\log t}\right) \leq \hat{C}|A|t\exp\{-\frac{t}{(\log t)^3}\}.$$

**Proof 3** *We denote the following event,*

$$E_{r_k} = \{\| r(s',.) - \hat{r}_k(s',.) \|_\infty < \frac{2\sigma}{\log(2+k)}\}.$$

*Thus, conditioned on the event $\bigcap_{i=1}^t E_{r_t}$ and for $t \geq 4$, we bound $|\hat{N}_t(s,a) - N_t^*(s,a)|$ as*

$$|\hat{N}_t(s,a) - N_t^*(s,a)| \leq \sum_{k=1}^t |\hat{\pi}_k(a|s) - \pi^*(a|s)| + \sum_{k=1}^t \lambda_k$$

$$\leq \sum_{k=1}^t \| \hat{\pi}_k(.|s) - \pi^*(.|s) \|_\infty + \sum_{k=1}^t \lambda_k$$

$$\leq \sum_{k=1}^t \| \hat{\pi}_k(.|s) - \pi^*(.|s) \|_p + \sum_{k=1}^t \lambda_k$$

$$\leq L \sum_{k=1}^t \| \hat{Q}_k(s',.) - Q(s',.) \|_p + \sum_{k=1}^t \lambda_k (Lemma\ 2)$$

$$\leq L|A|^{\frac{1}{p}} \sum_{k=1}^t \| \hat{Q}_k(s',.) - Q(s',.) \|_\infty + \sum_{k=1}^t \lambda_k (\ Property\ of\ p\text{-}norm)$$

$$\leq L|A|^{\frac{1}{p}}\gamma^d \sum_{k=1}^t \| \hat{r}_k(s'',.) - r(s'',.) \|_\infty + \sum_{k=1}^t \lambda_k (Contraction\ 3.1)$$

$$\leq L|A|^{\frac{1}{p}}\gamma^d \sum_{k=1}^t \frac{2\sigma}{\log(2+k)} + \sum_{k=1}^t \lambda_k$$

$$\leq L|A|^{\frac{1}{p}}\gamma^d \int_{k=0}^t \frac{2\sigma}{\log(2+k)}dk + \int_{k=0}^t \frac{|A|}{\log(1+k)}dk$$

$$\leq \frac{Ct}{\log t}.$$

*for some constant $C$ depending on $|A|, p, d, \sigma, L$, and $\gamma$. Finally,*

$$\mathbb{P}(|\hat{N}_t(s,a) - N_t^*(s,a)| \geq \frac{Ct}{\log t}) \leq \sum_{i=1}^{t} \mathbb{P}(E_{r_t}^c) = \sum_{i=1}^{t} 4|A| \exp(-\frac{t}{(\log(2+t))^3})$$

$$\leq 4|A|t \exp(-\frac{t}{(\log(2+t))^3})$$

$$= O(t \exp(-\frac{t}{(\log(t))^3})).$$

**Lemma 4** *Consider the E3W policy applied to a tree. At any node $s$ of the tree, Let us define $N_t^*(s,a) = \pi^*(a|s).t$, and $N_t(s,a)$ as the number of times action $a$ have been chosen until time step $t$. There exists some $C$ and $\hat{C}$ such that*

$$\mathbb{P}(|N_t(s,a) - N_t^*(s,a)| > \frac{Ct}{\log t}) \leq \hat{C}t \exp\{-\frac{t}{(\log t)^3}\}.$$

**Proof 4** *Based on the result from Lemma 3, we have*

$$\mathbb{P}(|N_t(s,a) - N_t^*(s,a)| > (1+C)\frac{t}{\log t}) \leq Ct \exp\{-\frac{t}{(\log t)^3}\}$$

$$\leq \mathbb{P}(|\hat{N}_t(s,a) - N_t^*(s,a)| > \frac{Ct}{\log t}) + \mathbb{P}(|N_t(s,a) - \hat{N}_t(s,a)| > \frac{t}{\log t})$$

$$\leq 4|A|t \exp\{-\frac{t}{(\log(2+t))^3}\} + 2|A| \exp\{-\frac{t}{(\log(2+t))^2}\} (\textit{Lemma 3 and (16)})$$

$$\leq O(t \exp(-\frac{t}{(\log t)^3})).$$

**Theorem 1** *At the root node $s$ of the tree, defining $N(s)$ as the number of visitations and $V_{\Omega^*}(s)$ as the estimated value at node $s$, for $\epsilon > 0$, we have*

$$\mathbb{P}(|V_\Omega(s) - V_\Omega^*(s)| > \epsilon) \leq C \exp\{-\frac{N(s)\epsilon}{\hat{C}(\log(2+N(s)))^2}\}.$$

**Proof 5** *We prove this concentration inequality by induction. When the depth of the tree is $D = 1$, from Proposition 1, we get*

$$|V_\Omega(s) - V_\Omega^*(s)| = \| \Omega^*(Q_\Omega(s,.)) - \Omega^*(Q_\Omega^*(s,.)) \|_\infty \leq \gamma \| \hat{r} - r^* \|_\infty \, (\textit{Contraction})$$

*where $\hat{r}$ is the average rewards and $r^*$ is the mean reward. So that*

$$\mathbb{P}(|V_\Omega(s) - V_\Omega^*(s)| > \epsilon) \leq \mathbb{P}(\gamma \| \hat{r} - r^* \|_\infty > \epsilon).$$

*From Lemma 1, with $\epsilon = \frac{2\sigma\gamma}{\log(2+N(s))}$, we have*

$$\mathbb{P}(|V_\Omega(s) - V_\Omega^*(s)| > \epsilon) \leq \mathbb{P}(\gamma \| \hat{r} - r^* \|_\infty > \epsilon) \leq 4|A| \exp\{-\frac{N(s)\epsilon}{2\sigma\gamma(\log(2+N(s)))^2}\}$$

$$= C \exp\{-\frac{N(s)\epsilon}{\hat{C}(\log(2+N(s)))^2}\}.$$

*Let assume we have the concentration bound at the depth $D - 1$, Let us define $V_\Omega(s_a) = Q_\Omega(s,a)$, where $s_a$ is the state reached taking action $a$ from state $s$. then at depth $D - 1$*

$$\mathbb{P}(|V_\Omega(s_a) - V_\Omega^*(s_a)| > \epsilon) \leq C \exp\{-\frac{N(s_a)\epsilon}{\hat{C}(\log(2+N(s_a)))^2}\}. \tag{17}$$

*Now at the depth $D$, because of the Contraction Property, we have*

$$|V_\Omega(s) - V_\Omega^*(s)| \leq \gamma \| Q_\Omega(s,.) - Q_\Omega^*(s,.) \|_\infty$$

$$= \gamma |Q_\Omega(s,a) - Q_\Omega^*(s,a)|.$$

*So that*

$$\mathbb{P}(|V_\Omega(s) - V_\Omega^*(s)| > \epsilon) \leq \mathbb{P}(\gamma \parallel Q_\Omega(s,a) - Q_\Omega^*(s,a) \parallel > \epsilon)$$

$$\leq C_a \exp\{-\frac{N(s_a)\epsilon}{\hat{C}_a(\log(2 + N(s_a)))^2}\}$$

$$\leq C_a \exp\{-\frac{N(s_a)\epsilon}{\hat{C}_a(\log(2 + N(s)))^2}\}.$$

*From (17), we can have* $\lim_{t\to\infty} N(s_a) = \infty$ *because if* $\exists L, N(s_a) < L$, *we can find* $\epsilon > 0$ *for which (17) is not satisfied. From Lemma 4, when $N(s)$ is large enough, we have* $N(s_a) \to \pi^*(a|s)N(s)$ *(for example* $N(s_a) > \frac{1}{2}\pi^*(a|s)N(s)$), *that means we can find $C$ and $\hat{C}$ that satisfy*

$$\mathbb{P}(|V_\Omega(s) - V_\Omega^*(s)| > \epsilon) \leq C \exp\{-\frac{N(s)\epsilon}{\hat{C}(\log(2 + N(s)))^2}\}.$$

**Lemma 5** *At any node $s$ of the tree, $N(s)$ is the number of visitations. We define the event*

$$E_s = \{\forall\, a \text{ in } \mathcal{A}, |N(s,a) - N^*(s,a)| < \frac{N^*(s,a)}{2}\} \text{ where } N^*(s,a) = \pi^*(a|s)N(s),$$

*where $\epsilon > 0$ and $V_{\Omega^*}(s)$ is the estimated value at node $s$. We have*

$$\mathbb{P}(|V_\Omega(s) - V_\Omega^*(s)| > \epsilon|E_s) \leq C \exp\{-\frac{N(s)\epsilon}{\hat{C}(\log(2 + N(s)))^2}\}.$$

**Proof 6** *The proof is the same as in Theorem 2. We prove the concentration inequality by induction. When the depth of the tree is $D = 1$, from Proposition 1, we get*

$$|V_\Omega(s) - V_\Omega^*(s)| = \parallel \Omega^*(Q_\Omega(s,.)) - \Omega^*(Q_\Omega^*(s,.)) \parallel \leq \gamma \parallel \hat{r} - r^* \parallel_\infty \text{ (Contraction Property)}$$

*where $\hat{r}$ is the average rewards and $r^*$ is the mean rewards. So that*

$$\mathbb{P}(|V_\Omega(s) - V_\Omega^*(s)| > \epsilon) \leq \mathbb{P}(\gamma \parallel \hat{r} - r^* \parallel_\infty > \epsilon).$$

*From Lemma 1, with $\epsilon = \frac{2\sigma\gamma}{\log(2+N(s))}$ and given $E_s$, we have*

$$\mathbb{P}(|V_\Omega(s) - V_\Omega^*(s)| > \epsilon) \leq \mathbb{P}(\gamma \parallel \hat{r} - r^* \parallel_\infty > \epsilon) \leq 4|A| \exp\{-\frac{N(s)\epsilon}{2\sigma\gamma(\log(2 + N(s)))^2}\}$$

$$= C \exp\{-\frac{N(s)\epsilon}{\hat{C}(\log(2 + N(s)))^2}\}.$$

*Let assume we have the concentration bound at the depth $D - 1$, Let us define $V_\Omega(s_a) = Q_\Omega(s,a)$, where $s_a$ is the state reached taking action $a$ from state $s$, then at depth $D - 1$*

$$\mathbb{P}(|V_\Omega(s_a) - V_\Omega^*(s_a)| > \epsilon) \leq C \exp\{-\frac{N(s_a)\epsilon}{\hat{C}(\log(2 + N(s_a)))^2}\}.$$

*Now at depth $D$, because of the Contraction Property and given $E_s$, we have*

$$|V_\Omega(s) - V_\Omega^*(s)| \leq \gamma \parallel Q_\Omega(s,.) - Q_\Omega^*(s,.) \parallel_\infty$$

$$= \gamma|Q_\Omega(s,a) - Q_\Omega^*(s,a)|(\exists a, \text{ satisfied}).$$

*So that*

$$\mathbb{P}(|V_\Omega(s) - V_\Omega^*(s)| > \epsilon) \leq \mathbb{P}(\gamma \parallel Q_\Omega(s,a) - Q_\Omega^*(s,a) \parallel > \epsilon)$$

$$\leq C_a \exp\{-\frac{N(s_a)\epsilon}{\hat{C}_a(\log(2 + N(s_a)))^2}\}$$

$$\leq C_a \exp\{-\frac{N(s_a)\epsilon}{\hat{C}_a(\log(2 + N(s)))^2}\}$$

$$\leq C \exp\{-\frac{N(s)\epsilon}{\hat{C}(\log(2 + N(s)))^2}\}(\text{because of } E_s)$$

.

**Theorem 2** *Let $a_t$ be the action returned by algorithm E3W at iteration $t$. Then for $t$ large enough, with some constants $C, \hat{C}$,*

$$\mathbb{P}(a_t \neq a^*) \leq Ct \exp\{-\frac{t}{\hat{C}\sigma(\log(t))^3}\}.$$

**Proof 7** *Let us define event $E_s$ as in Lemma 5. Let $a^*$ be the action with largest value estimate at the root node state $s$. The probability that E3W selects a sub-optimal arm at $s$ is*

$$\mathbb{P}(a_t \neq a^*) \leq \sum_a \mathbb{P}(V_\Omega(s_a)) > V_\Omega(s_{a^*})|E_s) + \mathbb{P}(E_s^c)$$

$$= \sum_a \mathbb{P}((V_\Omega(s_a) - V_\Omega^*(s_a)) - (V_\Omega(s_{a^*}) - V_\Omega^*(s_{a^*})) \geq V_\Omega^*(s_{a^*}) - V_\Omega^*(s_a)|E_s) + \mathbb{P}(E_s^c).$$

*Let us define $\Delta = V_\Omega^*(s_{a^*}) - V_\Omega^*(s_a)$, therefore for $\Delta > 0$, we have*

$$\mathbb{P}(a_t \neq a^*) \leq \sum_a \mathbb{P}((V_\Omega(s_a) - V_\Omega^*(s_a)) - (V_\Omega(s_{a^*}) - V_\Omega^*(s_{a^*})) \geq \Delta|E_s) + +\mathbb{P}(E_s^c)$$

$$\leq \sum_a \mathbb{P}(|V_\Omega(s_a) - V_\Omega^*(s_a)| \geq \alpha\Delta|E_s) + \mathbb{P}(|V_\Omega(s_{a^*}) - V_\Omega^*(s_{a^*})| \geq \beta\Delta|E_s) + \mathbb{P}(E_s^c)$$

$$\leq \sum_a C_a \exp\{-\frac{N(s)(\alpha\Delta)}{\hat{C}_a(\log(2 + N(s)))^2}\} + C_{a^*} \exp\{-\frac{N(s)(\beta\Delta)}{\hat{C}_{a^*}(\log(2 + N(s)))^2}\} + \mathbb{P}(E_s^c),$$

*where $\alpha + \beta = 1$, $\alpha > 0$, $\beta > 0$, and $N(s)$ is the number of visitations the root node $s$. Let us define $\frac{1}{\hat{C}} = \min\{\frac{(\alpha\Delta)}{C_a}, \frac{(\beta\Delta)}{C_{a^*}}\}$, and $C = \frac{1}{|A|} \max\{C_a, C_{a^*}\}$ we have*

$$\mathbb{P}(a \neq a^*) \leq C \exp\{-\frac{t}{\hat{C}\sigma(\log(2 + t))^2}\} + \mathbb{P}(E_s^c).$$

*From Lemma 4, $\exists C', \hat{C}'$ for which*

$$\mathbb{P}(E_s^c) \leq C't \exp\{-\frac{t}{\hat{C}'(\log(t))^3}\},$$

*so that*

$$\mathbb{P}(a \neq a^*) \leq O(t \exp\{-\frac{t}{(\log(t))^3}\}).$$

**Theorem 3** *Consider an E3W policy applied to the tree. Let $\kappa_i = \nabla\Omega^*(a_i|s) + \frac{L}{p}\sqrt{\hat{C}\sigma^2 \log \frac{C}{\delta}/2n}$, $\chi_i = \nabla\Omega^*(a_i|s) - \frac{L}{p}\sqrt{\hat{C}\sigma^2 \log \frac{C}{\delta}/2n}$, where $\nabla\Omega^*(.|s)$ is the policy with respect to the mean value vector $V(\cdot)$ at the root node $s$. For any $\delta > 0$, with probability at least $1 - \delta$, $\exists$ constant $L, p, C, \hat{C}$ so that the pseudo regret $R_n$ satisfies*

$$nV^* - n\sum_i V_i\Big(\kappa_i + \frac{L}{p}\big(\frac{\tau(U_\Omega - L_\Omega)}{1 - \gamma}\big)\Big) \leq R_n \leq nV^* - n\sum_i V_i\Big(\chi_i - \frac{L}{p}\big(\frac{\tau(U_\Omega - L_\Omega)}{1 - \gamma}\big)\Big).$$

**Proof 8** *From Lemma 2 given two policies $\pi^{(1)} = \nabla\Omega^*(r^{(1)})$ and $\pi^{(2)} = \nabla\Omega^*(r^{(2)})$, $\exists L$, such that*

$$\| \pi^{(1)} - \pi^{(2)} \|_p \leq L \| r^{(1)} - r^{(2)} \|_p \leq L\frac{1}{p} \| r^{(1)} - r^{(2)} \|_\infty.$$

*From (13), we have the regret*

$$R_n = nV^* - \sum_i V_i \sum_{t=1}^n \hat{\pi}_t(a_i|s), \tag{18}$$

*where $\hat{\pi}_t(\cdot)$ is the policy at time step $t$, and $\mathbb{I}(\cdot)$ is the indicator function. $V^*$ is the optimal branch at the root node, $V_i$ is the mean value function of the branch with respect to action $i$, $V(\cdot)$ is the $|A|$*

*vector of value function at the root node. $\hat{V}(\cdot)$ is the $|A|$ estimation vector of value function at the root node. $\pi(.|s) = \nabla\Omega^*(V(\cdot))$ is the policy with respect to the $V(\cdot)$ vector at the root node.*

*Then for any $\delta > 0$, with probability at least $1 - \delta$, we have*

$$|\pi(a_i|s) - \hat{\pi}_t(a_i|s)| \leq \| \pi(.|s) - \hat{\pi}_t(.|s) \|_\infty \leq \frac{L}{p} \| V(\cdot) - \hat{V}(\cdot) \|_\infty \ (\textit{Lemma 2}) \qquad (19)$$

$$\leq \frac{L}{p}|V(\cdot) - \hat{V}(\cdot)| \leq \frac{L}{p}\left( \frac{\tau(U_\Omega - L_\Omega)}{1 - \delta} + \sqrt{\frac{\hat{C}\sigma^2 \log \frac{C}{\delta}}{2N(s)}} \right)(\textit{Theorem 4})$$

*So that*

$$\pi(a_i|s) - \frac{L}{p}\left( \frac{\tau(U_\Omega - L_\Omega)}{1 - \delta} + \sqrt{\frac{\hat{C}\sigma^2 \log \frac{C}{\delta}}{2N(s)}} \right) \leq \hat{\pi}_t(a_i|s) \leq \pi(a_i|s) + \frac{L}{p}\left( \frac{\tau(U_\Omega - L_\Omega)}{1 - \delta} + \sqrt{\frac{\hat{C}\sigma^2 \log \frac{C}{\delta}}{2N(s)}} \right)$$

*so that*

$$R_n = nV^* - \sum_i V_i \sum_{t=1}^n \hat{\pi}_t(a_i|s) \leq nV^* - \sum_i V_i \sum_{t=1}^n \left( \pi(a_i|s) - \frac{L}{p}\Big(\frac{\tau(U_\Omega - L_\Omega)}{1 - \delta} + \sqrt{\frac{\hat{C}\sigma^2 \log \frac{C}{\delta}}{2n}}\Big) \right)$$

$$R_n \leq nV^* - \sum_i V_i \sum_{t=1}^n \left( \pi(a_i|s) - \frac{L}{p}\Big(\frac{\tau(U_\Omega - L_\Omega)}{1 - \delta} + \sqrt{\frac{\hat{C}\sigma^2 \log \frac{C}{\delta}}{2n}}\Big) \right)$$

$$R_n \leq nV^* - n\sum_i V_i \left( \pi(a_i|s) - \frac{L}{p}\Big(\frac{\tau(U_\Omega - L_\Omega)}{1 - \delta} + \sqrt{\frac{\hat{C}\sigma^2 \log \frac{C}{\delta}}{2n}}\Big) \right)$$

$$(20)$$

*And*

$$R_n \geq nV^* - \sum_i V_i \sum_{t=1}^n \left( \pi(a_i|s) + \frac{L}{p}\Big(\frac{\tau(U_\Omega - L_\Omega)}{1 - \delta} + \sqrt{\frac{\hat{C}\sigma^2 \log \frac{C}{\delta}}{2n}}\Big) \right)$$

$$R_n \geq nV^* - n\sum_i V_i \left( \pi(a_i|s) + \frac{L}{p}\Big(\frac{\tau(U_\Omega - L_\Omega)}{1 - \delta} + \sqrt{\frac{\hat{C}\sigma^2 \log \frac{C}{\delta}}{2n}}\Big) \right)$$

*In case of Maximum Entropy and Relative Entropy $p = 1$, because*

$$\| \pi^{(1)} - \pi^{(2)} \|_\infty \leq L \| r^{(1)} - r^{(2)} \|_\infty .$$

*So that we have for MENTS*

$$nV^* - n\sum_i V_i \Big(\kappa_i + L\big(\frac{\tau \log |A|}{1 - \gamma}\big)\Big) \leq R_n \leq nV^* - n\sum_i V_i \Big(\chi_i - L\big(\frac{\tau \log |A|}{1 - \gamma}\big)\Big).$$

*For RENTS, we have*

$$nV^* - n\sum_i V_i \Big(\kappa_i + L\big(\frac{\tau(\log |A| - \frac{1}{m})}{1 - \gamma}\big)\Big) \leq R_n \leq nV^* - n\sum_i V_i \Big(\chi_i - L\big(\frac{\tau(\log |A| - \frac{1}{m})}{1 - \gamma}\big)\Big)$$

*where $m = \min_a \pi(a|s)$.*
*In case of Tsallis Entropy $p = 2$ ( Niculae & Blondel (2017)), so that*

$$nV^* - n\sum_i V_i \Big(\kappa_i + \frac{L}{2}\big(\frac{|A| - 1}{2|A|}\frac{\tau}{1 - \gamma}\big)\Big) \leq R_n \leq nV^* - n\sum_i V_i \Big(\chi_i - \frac{L}{2}\big(\frac{|A| - 1}{2|A|}\frac{\tau}{1 - \gamma}\big)\Big)$$

Before derive the next theorem, we state here the Theorem 2 in Geist et al. (2019)

- Boundedness: for two constants $L_\Omega$ and $U_\Omega$ such that for all $\pi \in \Pi$, we have $L_\Omega \leq \Omega(\pi) \leq U_\Omega$, then

$$V^*(s) - \frac{\tau(U_\Omega - L_\Omega)}{1 - \gamma} \leq V^*_\Omega(s) \leq V^*(s). \qquad (21)$$

Where $\tau$ is the temperature and $\gamma$ is the discount constant.

**Theorem 4** *For any $\delta > 0$, with probability at least $1 - \delta$, the $\varepsilon_\Omega$ satisfies*

$$-\sqrt{\frac{\hat{C}\sigma^2 \log \frac{C}{\delta}}{2N(s)}} - \frac{\tau(U_\Omega - L_\Omega)}{1-\gamma} \leq \varepsilon_\Omega \leq \sqrt{\frac{\hat{C}\sigma^2 \log \frac{C}{\delta}}{2N(s)}}.$$

**Proof 9** *From Theorem 2, let us define* $\delta = C \exp\{-\frac{2N(s)\epsilon^2}{\hat{C}\sigma^2}\}$, *so that* $\epsilon = \sqrt{\frac{\hat{C}\sigma^2 \log \frac{C}{\delta}}{2N(s)}}$ *then for any* $\delta > 0$, *we have*

$$\mathbb{P}(|V_\Omega(s) - V_\Omega^*(s)| \leq \sqrt{\frac{\hat{C}\sigma^2 \log \frac{C}{\delta}}{2N(s)}}) \geq 1 - \delta.$$

*Then, for any $\delta > 0$, with probability at least $1 - \delta$, we have*

$$|V_\Omega(s) - V_\Omega^*(s)| \leq \sqrt{\frac{\hat{C}\sigma^2 \log \frac{C}{\delta}}{2N(s)}}$$

$$-\sqrt{\frac{\hat{C}\sigma^2 \log \frac{C}{\delta}}{2N(s)}} \leq V_\Omega(s) - V_\Omega^*(s) \leq \sqrt{\frac{\hat{C}\sigma^2 \log \frac{C}{\delta}}{2N(s)}}$$

$$-\sqrt{\frac{\hat{C}\sigma^2 \log \frac{C}{\delta}}{2N(s)}} + V_\Omega^*(s) \leq V_\Omega(s) \leq \sqrt{\frac{\hat{C}\sigma^2 \log \frac{C}{\delta}}{2N(s)}} + V_\Omega^*(s).$$

*From Proposition 1, we have*

$$-\sqrt{\frac{\hat{C}\sigma^2 \log \frac{C}{\delta}}{2N(s)}} + V^*(s) - \frac{\tau(U_\Omega - L_\Omega)}{1-\gamma} \leq V_\Omega(s) \leq \sqrt{\frac{\hat{C}\sigma^2 \log \frac{C}{\delta}}{2N(s)}} + V^*(s).$$

