# OpenReview forum: "Convex Regularization in Monte-Carlo Tree Search"
_ICLR.cc/2021/Conference — Reject_

### Official Review · AnonReviewer3 · 2020-10-28
**Nice results, but suspicious-looking details**

**Rating:** 5
**Confidence:** 3

**Review:**

The paper proposes a general framework for regularized Monte Carlo tree search, thereby generalizing the maximum entropy Monte Carlo planning of Xiao et al. The main result(s) looks very interesting, but some of the definitions and results don't seem right.

The main issue
=============

The main problem is Proposition 1 and the definition of the convex conjugate:
1) The definition of the convex conjugate in (2) differs significantly from the standard one (e.g., the definition in the two cited papers) due to the multiplier \tau. In fact, this \tau, unless set to 1, strongly questions the validity of (3) - and basically all the usual properties that are true for the standard definition of convex conjugates.
2) The contraction property in Proposition 1 also seems strange. It is well known that the Bellman operators are contractions, but the property claimed here is not true for the standard definition of the convex conjugate: if \Omega^* is a contraction, then \Omega should also be a contraction which, in turn, would imply \Omega^*=0 due to \Omega***=\Omega*. Finally, in case \Omega* is indeed a contraction, the contraction parameter should depend on \tau as well.

As Proposition 1 is essential to the main result, the above issues make the the correctness of the main result questionable.


Additional remarks
================

First of all, it would be nice to have a brief summary of how the analysis works and what the main idea is.

Section 3.2: the part of the trajectory discussed above (6) presumably corresponds to the selection part, not the simulation. Additionally, it is not clear how equation (7) was obtained. Please discuss it in more detail.

Regarding (10) and (11): please prove these equations or add a citation to some work where they are proved.

Equation (13): what is V_i? (Presumably, i_n stands for the action taken in step n.)

Regarding the experiments, they do show the superiority of proposed method on some benchmark tasks, such as CartPole, Acrobot and a couple of Atari games. However, it is not clear whether these tasks are the best to test UCT as the strength of UCT is the doing in-depth exploration on search trees with large branching factors, where the actions could have significant consequences in a very delayed fashion (in terms of reward).

---

> ### Author Response · Authors · 2020-11-16
> **Response to AnonReviewer3**
>
> We thank the reviewer for the effort to review our paper and the insightful feedback.
>
> Regarding the first issue, we remark that when $\Omega$ is strongly convex, $\tau \Omega$ is also strongly convex, thus all the properties shown in Proposition 1 are still correct (other works use the same formula, e.g. Equation (1) in Niculae and Blondel (2017)). In detail, the unique maximizing argument property, i.e. the policy maximizing the regularized objective, is still correct since $\tau \Omega$ is strongly convex.
>
> About the second issue, we clarify that the boundedness property is scaled with $\tau$ and the contraction property is guaranteed as shown in Proposition 2 in (Geist et al., 2019). Note that $\tau$ is simply a scaling factor that does not affect the property of contraction: for example, we can replace the original reward $r$ by $\frac{r}{\tau}$ and get the same formula of (Geist et al., 2019).
>
> With respect to the additional remarks:
>
> “First of all, it would be nice to have a brief summary of how the analysis works and what the main idea is.”: Our analysis aims at deriving the regularized counterparts of the convergence and regret theorems of UCT. While the exponential convergence rate to the regularized objective is already provided in Xiao et al (2019) for the case of maximum entropy, we demonstrate that all convex regularizers, e.g. relative and Tsallis entropy, enjoy this property. Our regret theorem is an extension of the UCT regret bound, and we provide both the standard form for a generic convex regularizer (Theorem 3), and the specialized ones for the considered entropies (Corollaries 1, 2, 3). This allows us to prove that the regret bound of TENTS is tighter in problems with many actions, while the regret of RENTS is tighter than the one of MENTS if a good prior is provided (for the contribution of the component $\frac{1}{m}$). Finally, since the regularized optimal value is different from the unregularized one, i.e. the one of UCT, we are interested in bounding the approximation error of these values. Like for the regret, Theorem 4 provides a bound for a generic convex regularizer, while Corollaries 4, 5, 6, specialize to each entropy regularizer. Again, Tsallis is beneficial in problems with many actions, while the prior of RENTS can make the bound tighter. We hope this clarifies the doubts of the reviewer. If not, we would be glad to be more helpful.
>
> Equation (6) corresponds indeed to the simulation phase, as stated in the paper. To be more clear, the selection phase does not involve the backup of value-functions, but only the exploration of the tree until an unvisited node is reached. On the other hand, the simulation results in a trajectory $s_0, a_0, s_1, a_1, \dots$ and in the update of the value of each node according to (6).
>
> Equation (7) is composed of two parts: the first part $\nabla \Omega^* (\frac{Q_\Omega(s_t)}{\tau})(a_t)$ is the regularized policy corresponding to the maximizing argument of the Legendre-Fenchel transform; the second part $\frac{1}{|\mathcal{A}|}$ corresponds to uniform sampling of actions. The parameter $\lambda$ trades-off exploitation of the regularized policy with uniform exploration, as done in E2W, and well-known $\varepsilon$-greedy policy in RL. This is convenient, although not necessary, to provide additional exploration to the one already induced by the regularized policy, still maintaining the same theoretical guarantees.
>
> Regarding (10) and (11): Thanks for pointing out the missing reference. We added the reference (Lee et al., 2018) to these two equations.
>
> Equation (13): Thanks for pointing out this problem of notation. We fixed it in the revised paper.
>
> Regarding the experiments:
>
> We hope that the empirical analysis of the toy problem added in the revised paper clarifies the doubts of the reviewer about the considered problems. These results show that UCT is still a valid choice for big trees, i.e. high branching factor and depth, being  the algorithm with the lowest regret, for k={12,16} and d={1, 4, 5}, and having faster convergence to the optimal value compared to RENTS and MENTS. Remarkably, our novel method TENTS seems to be the best choice, being almost always better, or at least comparable, to UCT.
>
> References:
>
> - Matthieu Geist,  Bruno Scherrer,  and Olivier Pietquin. A theory of regularized markov decision processes. In International Conference on Machine Learning (ICML), pp. 2160–2169, 2019.
> - Vlad Niculae, Mathieu Blondel. A Regularized Framework for Sparse and Structured Neural Attention. NIPS. 2017.

---

### Official Review · AnonReviewer4 · 2020-10-28
**Solid theoretical results, limited experimental results**

**Rating:** 5
**Confidence:** 4

**Review:**

##########################################################################

Summary:


The paper provides theoretical analysis of the regularized backup for MCTS. The paper carries out detailed analysis (regret, errors analysis) on three instantiations of the regularizations. Finally, the paper provides some empirical gains on certain toy domains and some atari games.

##########################################################################

Reasons for score:


Overall, I vote for rejection. I am not very familiar with the theoretical results of the MCTS literature - however, it seems that the idea of adding convex regularization is not new in rl literature overall. I cannot be a good judge for the theoretical contribution and I will focus more on the empirical side of the paper.

##########################################################################Pros:


1. Related work. The paper seems to miss some highly related literature, in particular:

[1] Buesing et al, Approximate Inference in Discrete Distributions with Monte Carlo Tree Search and Value Functions, AISTATS 2020

[2] Grill et al, Monte Carlo Tree Search as Regularized Policy Optimization, ICML 2020

[1] uses entropy-regularized MCTS backup; [2] relates MCTS to policy optimization with convex regularization. In particular, [2] proposes that the conventional MCTS backup used in Alpha-Zero (which is different from the max-MCTS backup), is carrying out approximate regularizations. I hope that the author could clarify on the connection between this work and these two pieces of prior work.

2. Regarding results in Table 2, I wonder how many seeds do the authors run per game per algorithmic baseline. Does each number correspond to a mean value across a few seeds, or it is just a single run? Could the author also clarify how the t-test is done to denote significant differences? I would expect such tests to be run on averages over a collection of seeds.

##########################################################################

Please address and clarify the concerns above.


#########################################################################

---

> ### Author Response · Authors · 2020-11-16
> **Response to AnonReviewer4**
>
> We thank the reviewer for the insightful review and for letting us know about the two recent research articles related to our work.
>
> “It seems that the idea of adding convex regularization is not new in rl literature overall”: the idea of using entropy regularization, which is a special case of convex regularization, in the Bellman equation has been used extensively in the Reinforcement Learning (RL) literature, resulting in well-known methods such as REPS, TRPO, PPO, COPOS, SAC. However, we provide the first study on the use of convex regularization in MCTS, which has only been studied, only for maximum entropy and mostly empirically, in MENTS (Xiao et al., 2019).
>
> To answer specifically about the two mentioned papers:
>
> (Buesing et al., AISTATS 2020) use MCTS as a component of an approximate inference approach. The proposed MCTS algorithm uses the same policy value backups of MENTS, but since the paper focuses on approximate inference, MCTS is not analyzed. Nevertheless, the paper shows the usefulness of entropy regularized MCTS methods, therefore we are citing it in the introduction of the revised paper.
>
> (Grill et al., ICML 2020) show that tree search in Alphazero is an approximate solution to reversed KL regularization between the prior and current policy. The proposed tree search policy is a special case of the tree search policy in our work. However, (Grill et al., ICML 2020) use the same value backup operator of UCT, which differs from our convex regularized operators, and, contrary to our work, (Grill et al., ICML 2020) do not provide theoretical study of the convergence, regret, and approximation error. Nevertheless, we think this paper is related to ours, thus we are citing it in the related works of the revised paper.
>
> Regarding the results in Table 2, we confirm that we ran 100 random seeds for each method on each Atari benchmark task. The numbers in the table correspond to the mean cumulative reward over 100 random seeds. In each benchmark task, we performed a t-test over all the seeds with p=0.95 to compare each method to the method with the highest mean.

---

### Official Review · AnonReviewer1 · 2020-10-29
**Official Blind Review #1**

**Rating:** 8
**Confidence:** 4

**Review:**

This paper generalizes and build on top of the MENTS/E2W, and shows that the entropy regularization can be replaced with any convex regularization. It uses the tools of convex conjugates and duality to derive the theoretical results and the algorithm/updates. Empirical results on Atari games confirm the value of policy regularization in MCTS.

Strong points:
 - Theory generalizes MENTS/E2W

- Experiments further support the value of regularized polices, showing that entropy is not the only thing that "works".

- Paper brings important and interesting insights into MCTS, which is potentially very impactfull.

- Previous work is (to my understanding) well cited.

- While the paper relies on non-trivial operations, it's well written.

- The resulting algorithms/updates are "easy" to implement.

Weak points:
- This paper is very much incremental to MENTS/E2W, one could say it "just generalizes" MENTS.

- Missing connection to previous results of policy/values dualities (please see additional feedback)

- The empirical results are not very exciting.

Reasons for score:
While the empirical results don't bring anything exciting (especially when contrasted to MENTS), they still bring interesting insights. It almost seems that any regularization is relevant. Furthermore, the presented theory/connection coming from the duality is important - I do not think this connection of duality was presented in the MENTS paper. Thus it's satisfying to know that this is where the "magic" of softmax and entropy as used in MENTS is coming from.  The reason I really like this paper is that it helps to build more intuition about the regularized policies in MCTS, all the while generalizing the underlying theory.

Additional feedback:
On high level, this paper essentially explores the duality of policies and corresponding (regularized) values. This idea/result/notion of duality between policy and value appears quite often (a quick example that comes to my mind being extensive form games but surely others), and I think few sentences on this would help the reader to feel "less surprised" about some of the presented derivations and techniques, and overall better place it in the context of previous relevant work. Reading this paper, one could think that the duality of policies/values is novel observation, while in general it's not.

---

> ### Author Response · Authors · 2020-11-16
> **Response to AnonReviewer1**
>
> We thank the reviewer for thoroughly reading the paper and we are glad about the positive feedback.
>
> “This paper is very much incremental to MENTS/E2W, one could say it "just generalizes" MENTS.”: while we understand that the proposed algorithms, i.e. RENTS and TENTS, share many similarities with MENTS, we remark that our paper goes deeper into the analysis of the use of convex regularization in MCTS. As the reviewer pointed out, the study of duality in RL is not new, but our work is the first to provide a theoretical ground about duality in MCTS. Moreover, our theoretical findings about regret and approximation error are novel contributions w.r.t. MENTS.
>
> “The empirical results are not very exciting.”: although TENTS achieves the highest scores in Atari, our experimental results in the submitted paper are indeed not conclusive about the superiority of one method over the other. However, we think that this is reasonable. The idea of the paper is not to establish which method is the best, because there is not; instead, we shed a light on the theoretical and empirical benefit of each regularizer, considering all of them equally important. Nevertheless, the new empirical analysis of the toy problem in the revised paper gives stronger evidence about the advantages of TENTS.
>
> About discussing duality: we agree that the notion of duality should be discussed better. The introduction and the related works section are now providing more information about that.

---

### Official Review · AnonReviewer2 · 2020-10-29
**Review: Convex Regularization in Monte-Carlo Tree Search**

**Rating:** 4
**Confidence:** 1

**Review:**

-Summary

The authors consider planning for Markov Decision Process. Precisely they study the benefit of convex regularization in Monte-Carlo Tree Search (MCTS). They generalize the E2W by xiao et al., 2019 by considering any strictly convex function as regularizer instead of the intial negative entropy. They provide a regret analysis of this algorithm named E3W and prove that EW3 converges at an exponential rate to the solution of the regularized objective function. Then they consider three particular instances MENTS with the Shannon entropy as a regularizer, RENTS with relative entropy to the previous policy as regularizer, and TENTS with the Tsallis entropy. They compare empirically these algorithms with PUCT as policy search in Alpha-go style MCTS on CartPol, Acrobot, and Atari games.


-Score justification/Main comments:
The setting is not clearly written and some key definitions are not introduced properly (see specific comments below). Given that it is hard to understand the main results.


-Detailed comments

P2: It is V^\pi(s) = \sum_{a} \pi(a|s) Q^\pi(s,a). Is the number of actions finite?

P3, (2): You mean \max_{\pi_s} \sum_{a} \pi(a|s) Q^\pi(s,a) -\tau \Omega(\pi_s) ? Because T_{\pi}Q is a function of (s,a) and I do not understand the notation T_{\pi_s}Q_s (and Proposition 1 will be wrong since in (4) the bound on the rewards should appear, e.g. take Q_s = 0).

P3, (4): absolute value for |\Omega^*(Q_1) - ....|

P4, (7): what is the link with (6)? Here, by the choice of \lambda you force the exploration as with \epsilon-greedy with is what UCT is trying to avoid.

P4, Theorem 1: Could you define the estimated value V_\Omage(s). But at the end we would like to be close to the true optimal value V^*, can you deduce a bound for |V_\Omega -V^*|.

P4, Theorem 2: I assume that a^* is the optimal value for the regularizd objective?

P4: In TRPO, Schulman et al, 2015, it is rather the reverse relative entropy than the relative entropy which is used.

P5, Table 1: there should be the temperature parameter \tau here.

P5: V_i is not defined, a_i either, it is a sum over i in which set? I do not understand your definition of the regret.

P5, Theorem 3: There is a sign issue in 13 since le left hand is greater than the right-hand. So the regret of E3W is linear?

P7, figure 1: Is it UCT or PUCT used for the experiments because in the main test you say it is PUCT? And I would not say that UCT is "clearly the worst" approach in Figure 1.

P9, appendix B: According to your setting the reward is deterministic. I do not understand. And it is an assumption no? Which leaf node?

P10: Could you define properly r(a) and \hat{r}(a)? Why there is a sum over k in the inequality below "therefore", is it a sum over a? And there is an issue with the parenthesis. I do not understand how you can control the term in exp(-1/(\log(2+...))^3) by  exp(-t/(\log(2+...))^3) in the last inequality.

---

> ### Author Response · Authors · 2020-11-16
> **Response to AnonReviewer2**
>
> We thank the reviewer for the effort to review our paper. In the following, we seek to solve all the concerns:
>
> - P2: The number of actions is finite as MCTS generally works with a discrete, finite, action space. In the revised paper, we define the action space $\mathcal{A}$ as a finite discrete action space.
> - P3, (2):  To define the Legendre-Fenchel transform, we adopt the notation in Geist et al. (2019). As defined $Q_s$ and $\pi_s$ are respectively $Q(s,\cdot)$ and $\pi(\cdot | s)$ , i.e. the $Q$-function and the policy for each action in state $s$. Therefore, the max operator is, for each state $s$, w.r.t. the policy maximizing the regularized objective.
> - P3 (4): we are not sure whether the reviewer refers to the contraction property in proposition 1. In that case, we remark that it is defined with respect to the $\infty$-norm, as shown in (Geist et al., ICML 2019).
> - P4, (7): as noted by the reviewer, the parameter $\lambda$ trades-off exploitation of the regularized policy with uniform exploration, as done in E2W, and  $\varepsilon$-greedy in RL. Although not fundamental, as $\lambda = 0$ would be still an effective choice, adding uniform exploration is convenient for speeding up performance. Moreover, note that $\lambda \to 0$ for infinite visitations to the considered node, therefore only the regularized policy is used in the limit.
> - P4, Theorem 1: The bound w.r.t. the true optimal value $|V_\Omega -V^*|$ is analyzed in detail in section 4.2, for each entropy regularizer.
> - P4, Theorem 2: $a^*$ is indeed the optimal action for the regularized objective. We clarified this in the revised paper.
> - P4:. Thanks for pointing this out. TRPO indeed uses the reverse KL; however, we decided to use the more classical forward KL, as used previously e.g. in REPS (Peters et al, 2012). Nevertheless, we will consider the use of reverse KL for a future empirical study.
> - P5, Table 1: thank you for pointing this out. We added $\tau$ in the revised paper.
> - P5: we apologise for the lack of clarity of the regret paragraph. We improved the presentation in the revised paper.
> - P5, Theorem 3: Thank you for spotting this typo. We fixed it in the revised paper.
> - P7, figure 1: It is PUCT, but we decided to keep UCT for simplicity. We agree that the reported sentence was too strong, we modified it in the revised paper.
> - P9: As specified in the beginning of Section B in the appendix, and as also studied in the original UCT paper (Kocsis et al, 2006), the reward distribution at the leaf node is $σ^2$-subgaussian, which is the only assumption we used.
> - P10: we are not sure about the issue identified by the reviewer. In Appendix B, we define that $\hat{r}$ and $r$ are respectively the average and the expected reward at the leaf node. Sum over $k$ here is with respect to the time step $k$. Regarding the term $\exp(\frac{-t}{\log(2+\dots)^3})$, it is actually upper bounded by $4|\mathcal{A}| \exp(\frac{-t}{\log(2+\dots)^3})$. Hope this helps, otherwise we would be glad to further clarify this.
>
> References:
> - Matthieu Geist,  Bruno Scherrer,  and Olivier Pietquin. A theory of regularized Markov decision processes. In International Conference on Machine Learning (ICML), pp. 2160–2169, 2019
> - Levente Kocsis, Csaba Szepesv´ari, Jan Willemson. Improved Monte-Carlo Search. 2006.
> - Relative Entropy Policy Search. Jan Peters, Katharina Mulling, Yasemin Altun. AAAI. 2012.

---

### Author Response · Authors · 2020-11-16
**Revised paper**

We thank all the reviewers for the effort put in reading the paper and their insightful comments.
We submitted a revised version of the paper including the following changes requested by the reviewers:
- typo corrections;
- improved notation;
- clarification of the connections of this work with the use of duality in RL.

Moreover, to provide more experimental results and evidence of the effect of our theoretical bounds in practice, we added an empirical analysis of the synthetic tree toy problem described in the MENTS paper (Xiao et al 2019). In this problem, we can tune the branching factor and the depth of a tree, and this allows us to assess how the performance of each regularizer are affected by the size of the tree. We observe that, as expected from our theoretical findings, TENTS is robust w.r.t. increasing branching factor, i.e. number of actions, having the fastest convergence rate to the regularizer optimal value. The error w.r.t. the unregularized optimal value is the smallest among the regularizer, as shown in Section 4.2, and the regret is generally not high as the one of RENTS and MENTS, as proven in Section 4.1. Overall, although we think that the superiority of each regularizer depends on the problem at hand (in Cartpole and Acrobot RENTS seems to be a, slightly more, valid choice), this toy experiment, together with the Atari one, provides further evidence of the effectiveness of Tsallis entropy as a convex regularizer in MCTS. An intuitive explanation of this, besides to our theoretical bounds, can be that sparse policies speed up the search of optimal actions over the tree, reducing the time spent exploring not promising branches.

---

### Decision · Program_Chairs · 2021-01-07
**Final Decision**

**Decision:**

Reject

**Comment:**

Most of the reviewers pointed out a lack of rigor of this submission, unclear contributions, not too convincing claims and empirical gains. I thank the authors for the effort put in revising the paper and responding to the reviewer concerns. However, the reviewers did not deem them convincing enough.

---

> ### Comment · ~Carlo_D'Eramo2 · 2021-01-14
> **No feedback from reviewers**
>
> We want to kindly express our disappointment for the poor reviewing process for our submission. The scores of the reviews are significantly unbalanced, considering the low confidence of two of the lowest scores, and the high confidence of the only very good score. We believe we addressed all concerns of the reviewer, and our revised submission adds more empirical evidence as requested. After our replies, we did not receive any feedback from the reviewers, not even the acknowledgment of having read our rebuttal. We believe that the unbalanced average score, and the average low confidence, of our submission should have motivated the reviewers and the meta-reviewer to provide us some feedback, especially considering the potential for interaction of OpenReview.